# High-performance photonic transformers for DC voltage conversion

Bo Zhao [1,3,4], Sid Assawaworrarit[1,4], Parthiban Santhanam [1], Meir Orenstein[1,2] & Shanhui Fan [1✉]

Direct current (DC) converters play an essential role in electronic circuits. Conventional high-efficiency DC voltage converters, especially step-up type, rely on switching operation, where energy is periodically stored within and released from inductors and/or capacitors connected in a variety of circuit topologies. Since these energy storage components, especially inductors, are fundamentally difficult to scale down, miniaturization of switching converters proves challenging. Furthermore, the resulting switching currents produce significant electromagnetic noise. To overcome the limitations of switching converters, photonic transformers, where voltage conversion is achieved through light emission and detection processes, have been demonstrated. However, the demonstrated efficiency is significantly below that of the switching converter. Here we perform a detailed balance analysis and show that with a monolithically integrated design that enables efficient photon transport, the photonic transformer can operate with a near-unity conversion efficiency and high voltage conversion ratio. We validate the theory with a transformer constructed with off-the-shelf discrete components. Our experiment showcases near noiseless operation and a voltage conversion ratio that is significantly higher than obtained in previous photonic transformers. Our findings point to the possibility of a high-performance optical solution to miniaturizing DC power converters and improving the electromagnetic compatibility and quality of electrical power.

[1] Department of Electrical Engineering, Stanford University, Stanford, CA, USA. [2] Department of Electrical Engineering, Technion Israel Institute of Technology, Haifa, Israel. [3] Present address: Department of Mechanical Engineering, University of Houston, Houston, TX, USA. [4] These authors contributed equally: Bo Zhao, Sid Assawaworrarit. ✉email: shanhui@stanford.edu

Voltage conversion plays a critical role in electrical and electronic systems, bridging the gap between the voltage requirements of the power source/generation, distribution, and individual loads on the circuit. As a well-known example that underpins electrical power distribution networks, a voltage transformer (Fig. 1a) converts one alternating current (AC) voltage to another using the principle of magnetic induction. Such a transformer, however, cannot be applied directly for direct current (DC) voltage conversion.

Since many electronic devices rely on DC power, direct current (DC) voltage converters are of essential importance in electronics. The standard approach to DC voltage conversion, especially the step-up type, relies on switching converters (Fig. 1b). The core building blocks of these converters are intermediate energy storage elements (inductors and capacitors), and switches (e.g., transistors and diodes) that are temporally modulated to charge and discharge the energy storage elements[1]. Due to the difficulty of scaling down high-quality inductors[2], switching converters usually take up substantial real estate on-chip[3] or on the circuit board[4] and therefore represent a major obstacle in miniaturizing electronic devices[5]. Moreover, the switching action inevitably produces fluctuating internal voltages and currents, resulting in significant electromagnetic[6,7] or even acoustic noise[8].

Recently, photon-mediated voltage-conversion techniques have been proposed as an alternative solution for DC voltage conversion[9,10] to overcome some of the limitations of standard switching converters. Similar to a magnetic transformer, this technique naturally provides electric isolation between the input and output ports. Therefore, in this paper, we refer to such a photon-based voltage converter as a photonic transformer. While the electromagnetic field of the photons are time-varying, the frequencies of such variations are several orders of magnitude larger than what one can detect using electronic circuits. As a result, a photonic transformer operates as a DC device.

In a photonic transformer, a laser or a light-emitting diode (LED) is used to convert electrical energy to light energy, and either multiple photovoltaic (PV) cells laterally connected in series or a tandem multijunction PV cell that are used to convert photon flux back to electrical energy. In doing so, an input voltage that drives a laser or an LED can be boosted to a larger output voltage from the PV cells. Photonic transformers have been shown to produce no switching noise and are immune to the environment EMI and have been utilized to build DC voltage converters[10] and gate drivers[11]. The fact that the limiting efficiency of PV cells for converting near monochromatic light to electricity can approach unity[12], as well as the high theoretical efficiency of LEDs, make this technology attractive. However, the experimentally demonstrated efficiency, as well as voltage-conversion ratio, are still significantly below that of the standard switching voltage converter[10]. Therefore, it is important to establish a fundamental understanding of the performance potential of photonic transformers, and the practical pathways to reach such potential.

In this work, we provide a theoretical analysis of the fundamental performance potential for photonic transformers. Here, we focus on a photonic transformer that utilizes LEDs. From a fundamental thermodynamic point of view, the ultimate efficiency of an LED is higher than that of a laser. The electroluminescent efficiency of an LED, theoretically, can in fact exceed 100%[13,14], since an LED can operate as a heat engine that generates part of its light from the thermal energy of its environment[15–18]. In contrast, the efficiency of a laser has to be below 100%[19] since part of the electric pump power has to be converted to heat. In addition, the specific arrangement of the PV cells does not affect the performance of the photonic transformer in the ideal case. Therefore, our analysis focuses on using LED(s) facing laterally connected PV cells in series. Using a detailed balance analysis, we show that the efficiency of a photonic transformer can approach unity. The key to such high efficiency is to achieve highly efficient optical coupling between the LED and the PV cells. We thus propose a monolithic design where such a strong optical coupling between the LED and the PV cell can be achieved. To validate our theoretical model, we construct a photonic transformer prototype using discrete off-the-shelf components. Measurement results from the prototype are indeed in good agreement with the theoretical model once non-idealities in the circuit prototype are accounted for. Moreover, although the experimental setup as expected does not deliver high efficiency, it achieves a large voltage-conversion ratio (>40) that significantly exceeds what have been previously demonstrated in existing experiments on photonic transformers (<10)[10,11].

## Results

**Performance of ideal photonic transformers**. To derive the theoretical performance limit of our photonic transformer, we perform an analysis for the setup depicted in Fig. 1c with one

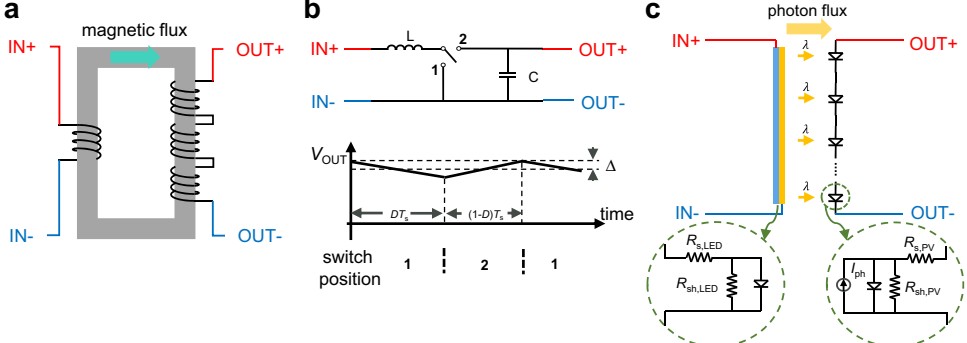

**Fig. 1 Schematic of an AC transformer, a step-up switching DC converter, and our proposed DC photonic transformer. a** Schematic of a conventional transformer. Input (AC) voltage on the primary winding generates an alternating magnetic flux in the magnetic core (gray rectangular ring), which induces a scaled version of the voltage on the secondary winding. The voltage-conversion ratio is determined by the winding ratio between the two coils. **b** Schematic of a typical switching step-up (boost) converter circuit and its steady-state output voltage as a function of time under linear-ripple approximation[1]. The circuit operates by temporally alternating the switch position between 1 and 2 resulting in the charging and discharging of the energy storage elements. $D$ duty cycle, $T_s$ switching period, $\Delta$ output voltage ripple magnitude. **c** Schematic of the proposed DC photonic transformer. The photon flux emitted by the LED and received by the PV cell acts as a current source for the external load of the PV cell. The inset shows the series resistance and the shunt resistance of the PV cell and LED.

LED facing $N$ identical PV cells. For planar LED and PV cells, when $N$ is large, we can reasonably assume the view factors from the LED to each PV cell, $f_{\text{LED}\to\text{PV}}$, are the same. In this case, each PV cell receives the same amount of illumination from the LED, and therefore generates the same voltage. We denote the voltage (current) of the LED and each PV cell as $V_{\text{LED}}$ ($I_{\text{LED}}$) and $V_{\text{PV}}$ ($I_{\text{PV}}$), respectively. Therefore, the input and output voltages are, respectively, $V_{\text{IN}} = V_{\text{LED}}$ and $V_{\text{OUT}} = NV_{\text{PV}}$, when the series resistance is negligible. The input and output currents are, respectively, $I_{\text{IN}} = I_{\text{LED}}$ and $I_{\text{OUT}} = -I_{PV}$. The voltage-conversion ratio ($W$) can be expressed as

$$W = \frac{V_{\text{OUT}}}{V_{\text{IN}}} = N \frac{V_{\text{PV}}}{V_{\text{LED}}} \tag{1}$$

And the power efficiency ($\eta$) can be obtained as

$$\eta = \frac{P_{\text{OUT}}}{P_{\text{IN}}} = \frac{V_{\text{OUT}}I_{\text{OUT}}}{V_{\text{IN}}I_{\text{IN}}} = N \frac{-I_{\text{PV}}V_{\text{PV}}}{I_{\text{LED}}V_{\text{LED}}} \tag{2}$$

Equation (1) indicates that the conversion ratio depends on how much voltage each PV cell can recover from the photon flux emitted by the LED. The power efficiency is a product of the voltage-conversion ratio and the current conversion ratio ($-I_{\text{PV}}/I_{\text{LED}}$). The voltage and current of the LED and each PV cell can be obtained from the detailed balance relations[20,21]:

$$[F_{\text{amb}\to\text{LED}} - F_{\text{LED}\to\text{amb}}] + N[F_{\text{PV}\to\text{LED}} - F_{\text{LED}\to\text{PV}}] - R_{\text{LED}}(V_{\text{LED}}) + \frac{I_{\text{LED}}}{q} = 0 \tag{3}$$

and

$$[F_{\text{amb}\to\text{PV}} - F_{\text{PV}\to\text{amb}}] + [F_{\text{LED}\to\text{PV}} - F_{\text{PV}\to\text{LED}}] - R_{\text{PV}}(V_{\text{PV}}) + \frac{I_{\text{PV}}}{q} = 0 \tag{4}$$

where $F_{a\to b}$ with $a, b =$ LED, PV, and amb (the ambient), is the photon flux emitted from object $a$ and absorbed by object $b$, $q$ is the elementary charge, $R$ is the total rate of nonradiative recombination. We use a sign convention such that a positive current flows from the p to the n region internally in each diode. Therefore, for the normal operation of the photonic transformer, $I_{\text{LED}} \geq 0$ and $I_{\text{PV}} \leq 0$. The ambient term includes all objects beyond the active regions of the LED and the PV, so the photon flux absorbed by the ambient represents the photon leakage from the converter, which should be minimized. The photon flux emitted by the ambient is much smaller compared to the photon flux emitted by the LED, and therefore can be neglected.

For the setup in the above panel of Fig. 1c, we first consider an ideal scenario where the LED and PV cells have the same bandgap with unity external quantum efficiency (EQE) above the bandgap, which implies that the nonradiative terms in Eqs. (3) and (4) are zero. In this case, we can model the emitted photon flux from object $a$ and absorbed by object $b$ as

$$F_{a\to b}(V_a) = A_a f_{a\to b} \int_{\omega_g}^{\infty} \frac{\omega^2}{4\pi^2 c^2} \frac{1}{\exp(\frac{\hbar\omega - qV_a}{kT}) - 1} d\omega \tag{5}$$

In Eq. (5), $a$ and $b$ denote the LED, PV, $\omega$ is the angular frequency, $\omega_g$ is the bandgap frequency, $c$ is the speed of light in vacuum, $k$ is the Boltzmann constant, $V_a$ is the voltage applied on object $a$, and $qV_a$ thus corresponds to the chemical potential of emitted photon[22]. $T$ is the temperature of the diode, and throughout the paper in the numerical calculations, we assume all objects are at room temperature with $T = 300\,\text{K}$. $A_a$ is the emitting surface area. $f_{a\to b}$ is the view factor from object $a$ to object $b$. If we further assume no photon leakage, then the terms in Eqs. (3) and (4) involving the ambient become zero, and we have $A_{\text{LED}} = NA_{\text{PV}}$, $f_{\text{LED}\to\text{PV}} = \frac{1}{N}$, and $f_{\text{PV}\to\text{LED}} = 1$. One

therefore could obtain $-NI_{\text{PV}} = I_{\text{LED}}$ based on these equations. Therefore, $W = N\eta$ based on Eqs. (1) and (2), and the maximum efficiency point is when $F_{\text{LED}\to\text{PV}} = F_{\text{PV}\to\text{LED}}$ based on Eqs. (3) and (4), i.e., $V_{\text{PV}} = V_{\text{LED}}$ based on Eq. (5). In other words, each PV cell at open-circuit condition will fully recover the voltage of the LED. In this case, the series-connected PV cell array is in open-circuit condition (i.e., $P_{\text{OUT}} = 0$) and outputs a boosted voltage that is $N$ times the input voltage according to Eq. (1). Therefore, the conversion ratio of the proposed photonic transformer depends on the number of PV cells. This dependence allows one to get in principle any desired high conversion ratio by selecting $N$.

The operation principle of photonic transformers is fundamentally different from that of conventional transformers or switching converters. The emission process in the LED and absorption process in the PV cell side are quantum processes[23]. In contrast, in the traditional transformers (Fig. 1a) or switching converters (Fig. 1b), the power exchange process can be described entirely classically.

**Performance of photonic transformers with nonidealities.** Now we analyze the expected performance of an actual photonic transformer. Based on the fluctuation–dissipation theorem (see "Methods"), taking into account the less-than-unity emissivity from the LED and absorptivity of the PV cell, the photon fluxes become

$$F_{\text{LED}\to\text{PV}}(V_{\text{LED}}) = \frac{A_{\text{LED}}f_{\text{LED}\to\text{PV}}}{8\pi^3} \int_{\omega_g}^{\infty} d\omega \iint \xi(\omega, k_x, k_y) \frac{\Theta(\omega, V_{\text{LED}})}{\hbar\omega} dk_x dk_y \tag{6}$$

and

$$F_{\text{PV}\to\text{LED}}(V_{\text{PV}}) = \frac{A_{\text{PV}}f_{\text{PV}\to\text{LED}}}{8\pi^3} \int_{\omega_g}^{\infty} d\omega \iint \xi(\omega, k_x, k_y) \frac{\Theta(\omega, V_{\text{PV}})}{\hbar\omega} dk_x dk_y \tag{7}$$

In the above two equations, $k_x$ and $k_y$ are the in-plane wavevectors, $\Theta$ is the photon energy in a mode at $\omega$

$$\Theta(\omega, T, V) = \frac{\hbar\omega}{\exp(\frac{\hbar\omega - qV}{kT}) - 1} \tag{8}$$

where $V$ is the bias on the active region and $qV$ acts as the chemical potential of photons[18,22], $A$ is the area, $f$ is the view factor, and $\xi(\omega, k_x, k_y)$ is the energy transmission coefficient summing over two polarizations and has a maximum value of 2. The various nonideality of the LEDs and the PV cells reduces the energy transmission coefficient to be <2. Moreover, with a finite load resistance, the operation point of the PV cells is shifted away from the open-circuit condition in order to have nonzero output power. Hence, the voltage recovery in general will not be complete. The voltage recovery will also be subject to penalties from the nonradiative processes and series resistances in both the LED and PV cells. Here, we show that a practical photonic transformer can still have excellent performance even under these considerations. As an example, we choose GaN (bandgap energy $\hbar\omega_g = 3.45\,\text{eV}$) as the active region material for both the LED and the PV cell since a high-performance GaN-based LED with 95% internal quantum efficiency has been experimentally demonstrated[24].

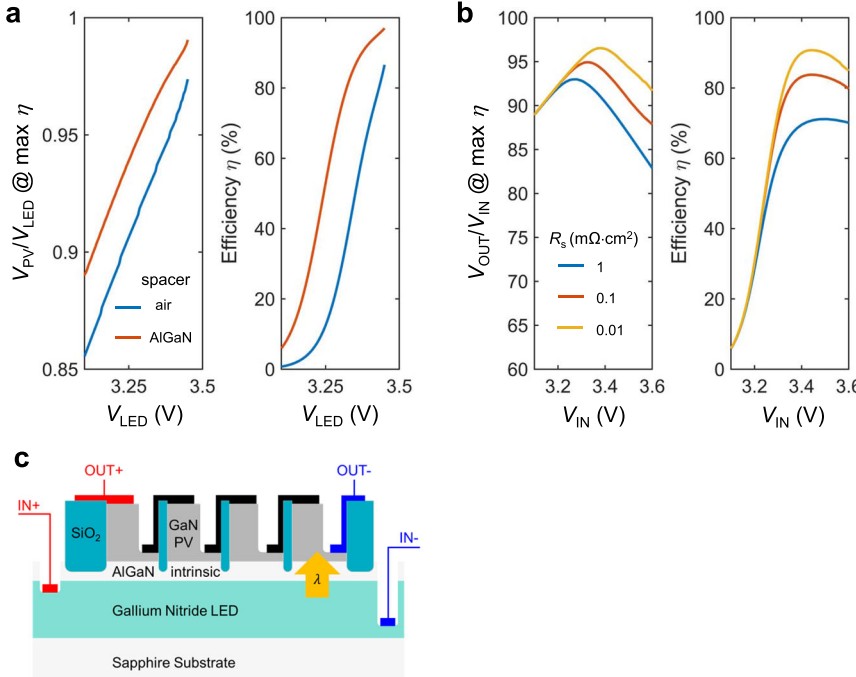

**Fig. 2 Theoretical performance of the proposed DC photonic transformer. a** Theoretical performance of the proposed GaN photonic transformer with the nonradiative recombination included ("Methods"). The light blue lines correspond to the case of a 1-μm thick air spacer layer between the LED and the PV cell, whereas the orange lines are for the case of an Al.5Ga.5N spacer layer with a thickness of 1 μm. Shown in the efficiency plot is the maximum possible efficiency by adjusting the load. **b** The performance of GaN photonic transformer with the Al.5Ga.5N spacer layer with different series resistance $R_s = R_{s,PV} = R_{s,LED}$. We assume the number of PV cells $N = 100$. **c** A conceptual monolithic device design of the proposed GaN photonic transformer with an intrinsic AlGaN spacer layer that provides electrical isolation and enhanced optical coupling.

We first consider the effect of nonradiative processes. The nonradiative recombination rates of the LED and PV cells are

$$R_a = A_a(C_{n,a}n_a + C_{p,a}p_a)(n_ap_a - n_{i,a}^2)t_a + \frac{1}{\tau_a}\frac{n_ap_a - n_{i,a}^2}{n_a + p_a + 2n_{i,a}}A_at_a \quad (9)$$

In the above equation, $a =$ PV cell or LED. The first and the second terms on the right-hand side are the Auger and Shockley–Read–Hall (SRH) recombination rates, respectively. $\tau$ is the bulk SRH lifetime, and $C_p$ and $C_n$ are the Auger recombination coefficients for holes and electrons, respectively. In the computation, we use the optical properties of GaN and AlGaN from ref. [25] and take into account the bias effect on the imaginary part of the dielectric function using the formula discussed in ref. [26]. We model nonradiative terms for both the LED and the PV cell using typical values $C_n = C_p = 5 \times 10^{-32}$ cm$^6$/s and $\tau = 20$ μs[27]. Using Eqs. (6), (7), and (9), together with Eqs. (3) and (4), we obtain a model of the I–V curve for both the LED and the PV cell. The I–V curves for the LED and the PV cell are coupled by the photonic flux. In the computation process, we start by fixing $V_{LED}$ to a specific value and then solve $I_{LED}$ and $I_{PV}$ at different $V_{PV}$. In this way, one can obtain the input power from the LED and the output power from the PV cell when the load on the PV cell is changing. The computation code for the photon fluxes can be accessed from ref. [28].

We next consider the effect of photonic exchange between the LED and the PV cell. For our device, it is critical to achieving efficient optical coupling between the active regions of the LED and the PV cell. This is similar to a conventional transformer in which one uses a ferromagnetic core to reduce leakage of magnetic energy. Equations (6) and (7) contain integration over

$k_x$ and $k_y$. In principle, there is no upper limit for $k_x$ and $k_y$ in such integral. In the case shown as blue curves in Fig. 2a, the LED and the PV cell are separated by an air spacer layer in the far-field regime with a thickness that is much larger than the emission wavelength (365 nm). In this case, only the propagating-wave channels in air with an in-plane wavevector $\beta = \sqrt{k_x^2 + k_y^2} < \omega/c$ are utilized for light extraction[29] from the front-emitting surface. Enhancement of coupling can be achieved if one can also utilize channels with $\beta > \omega/c$. A standard approach is to operate in the near-field regime where one reduces the thickness of the air spacer layer to be much smaller than the emission wavelength, so that channels with $\beta > \omega/c$, which are evanescent in air[30,31], can contribute. Maintaining such small thickness in the separation layer over a large area, however, represents a significant experimental challenge[32–38]. Instead, here we propose to use an Al.5Ga.5N layer as the spacer layer between the LED and the PV cell. Al.5Ga.5N is essentially transparent in the emission wavelength range of GaN and has a refractive index $n_{AlGaN}$ similar to that of GaN. In doing so, the channels with $\omega/c < \beta < n_{AlGaN}\omega/c$ are propagating in the AlGaN layer. As an illustration, we show the energy transmission coefficients for the far-field case with air and the case with AlGaN spacer layer in Fig. 3c, d, respectively. We see that in the far-field case with air, only the channels above the light line of air (green dashed line) contribute to photon transport. In contrast, for the case with the AlGaN layer, the channels between the light lines of GaN and air significantly contribute to the transport between the LED and the PV cell. The additional contribution of these channels greatly enhances the light-extraction efficiency of the LED such that the external quantum efficiency of the LED improves to 98.7%, as compared with that of 90.3% for the far-field case with an air spacer (Supplementary Fig. 1).

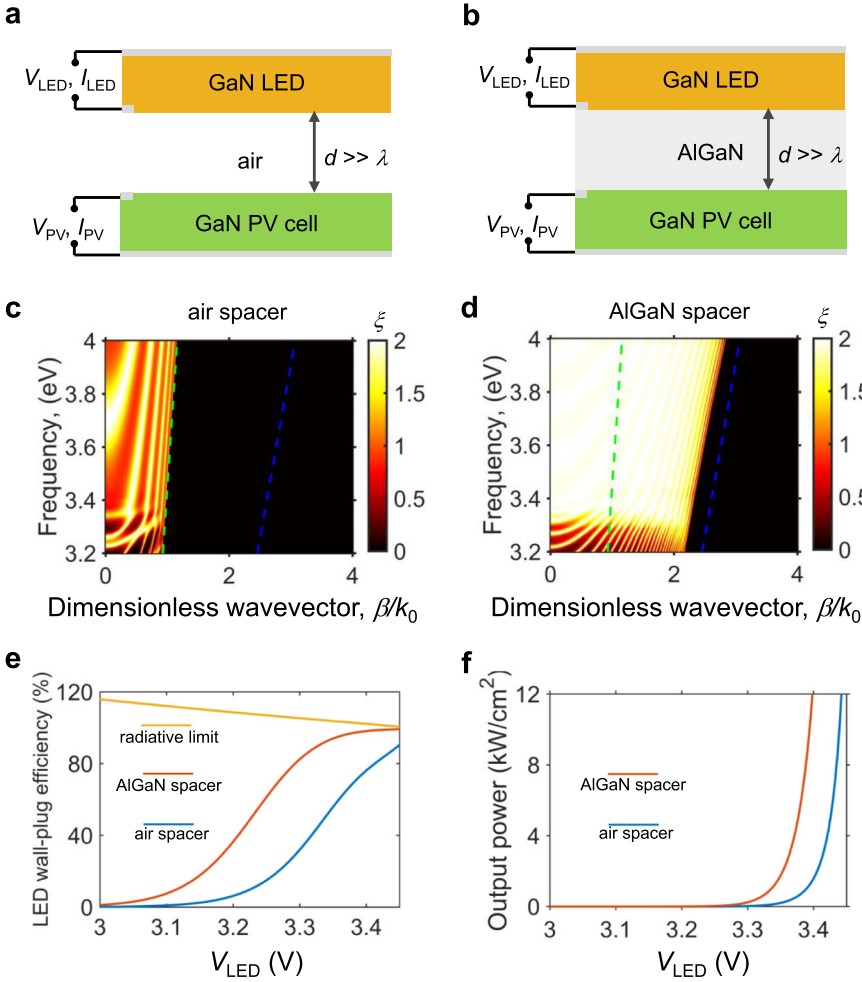

**Fig. 3 The impact of the AlGaN layer on optical coupling. a** The setup for the far-field case with an air spacer layer having a thickness of $d = 1\,\mu m$. **b** The setup for the AlGaN spacer layer case where the LED and the PV cell are coupled through a thin $Al_{.5}Ga_{.5}N$ layer with a thickness of $d = 1\,\mu m$. Metallic mirrors (the thick gray lines) are placed on the back of the LED and the PV cell for photon recycling and electrical contact purposes. They are modeled as a perfect conductor in the simulation. The thicknesses of the LED and the PV cell are set to be $1\,\mu m$ for both the air and the AlGaN spacer cases. **c, d** Photon transmission coefficients for the cases with air spacer layer and AlGaN spacer layer, respectively. The green dashed line is the light line of the vacuum given by $\beta = \omega/c$, and the blue dashed line is the light line of GaN given by $\beta = n_{GaN}\omega/c$, where $n_{GaN} = 2.65$ is the refractive index of GaN near its bandgap. The wavevectors are normalized by the free-space wavevector at the bandgap frequency. **e** The wall-plug efficiency of the LED in three cases: the radiative limit (i.e., nonradiative recombination rates are zero), the AlGaN spacer case, and the air spacer case. **f** Output power density of the GaN PV cell in the air spacer and AlGaN spacer cases at the maximum efficiency (power density) operation point. The wall-plug efficiency and the output power density are greatly enhanced in the AlGaN spacer case as compared to the far-field air spacer case.

The EQE of the PV cell is also greatly enhanced because of the index-matching AlGaN layer. The improvement in external quantum efficiency of the LED and the PV cells results in enhanced voltage-conversion ratio and conversion efficiency as indicated by the orange curves in Fig. 2a. Importantly, the enhancement here is not a near-field effect. The external quantum efficiency of the LED increases only by <1% if one reduces the AlGaN spacer thickness from $1\,\mu m$ to 10 nm. Thus, the thickness of the spacer can be larger than the wavelength without affecting the optical coupling. This is important in practice since the spacer layer also needs to provide electrical insulation between the LED and the PV cell.

In Fig. 3e, we show the wall-plug efficiency, i.e., the ratio of the emitted optical power and electrical power consumed by the LED, for three cases. The yellow curve corresponds to the theoretical upper bound in the radiative limit case where the nonradiative recombination rates are zero. As mentioned before, the wall-plug efficiency of the LED in the ideal case can exceed 100%[13,14]. The orange and the blue curves are for the cases with the AlGaN spacer and the air spacer, respectively, taking into account nonradiative recombination processes as discussed above. The wall-plug efficiency of the LED is greatly enhanced for the AlGaN spacer case, as compared with the far-field air spacer case. For example, the peak value of the wall-plug efficiency of the LED is enhanced from 90.6 to 99.3%. As a result, the overall conversion efficiency improves from 85 to ~97%. Thus the use of the AlGaN spacer layer allows us to approach the radiative limit in spite of the presence of nonradiative recombination in this system. The use of the AlGaN spacer layer can also greatly improve the power density of the photonic transformer compared to the air spacer case. As shown in Fig. 3f, with the AlGaN spacer layer, the maximum power density can reach $10\,kW/cm^2$ for an input voltage near 3.4 V, indicating that 1 W electrical power can be delivered in a footprint $\sim 0.01\,mm^2$. By comparison, a high-efficiency switching converter may have a volumetric power density[39] of 1 W per $10{-}100\,mm^3$. The use of a photonic transformer can therefore significantly reduce the device footprint required for voltage conversion.

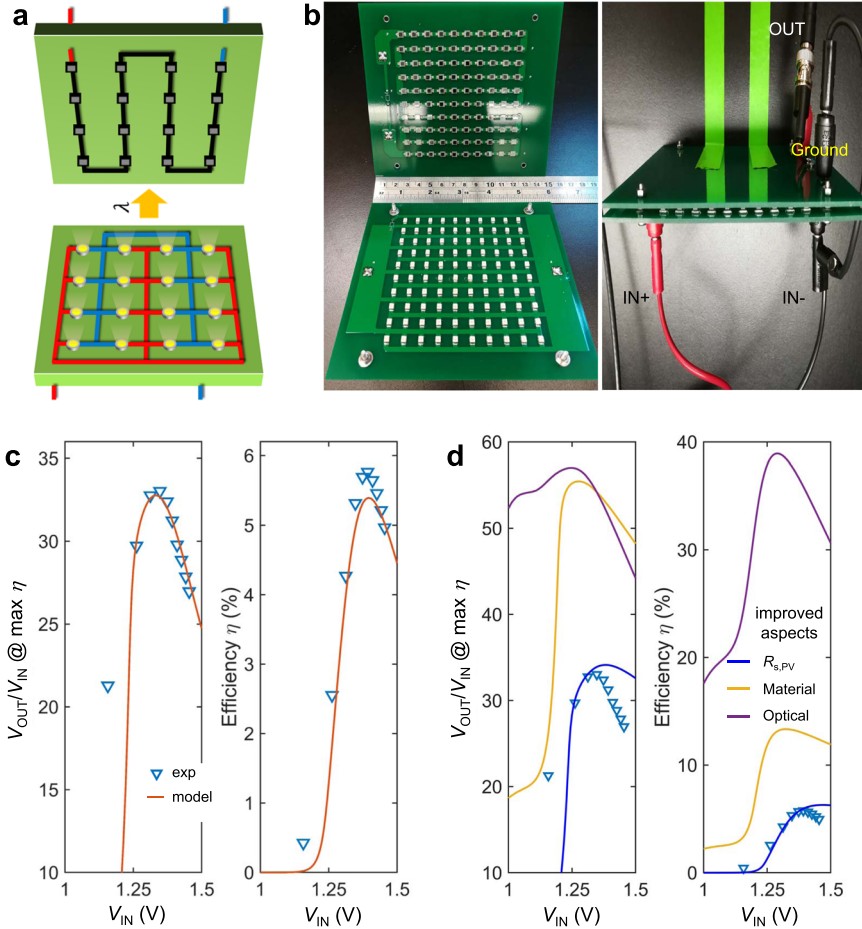

**Fig. 4 Proof-of-concept demonstration and verification of the device model of the DC-to-DC photonic transformer. a** Schematic of a board design with the same number of LEDs (bottom half) and PV cells (top half). **b** Photos of the two printed circuit boards consisting of LEDs and PV cells (left) and the assembled photonic transformer (right). **c** Measured (cyan triangles) voltage ratio and conversion efficiency. The red curves are predictions from the model. **d** Possible improvements on the photonic transformer prototype. The triangle data points are the measured data. In obtaining the curves in blue, we reduce the series resistance of the PV cells from 2.18 to 0.32 Ω cm². In obtaining the orange curves, we further reduce the nonradiative recombination terms for both the LED and the PV cell. For the purple curves, in addition to all the previous improvements, we set the light-extraction efficiency to be 80% and the view factor between the LED and the PV cell to be 1. The used parameters are listed in Supplementary Tables 1 and 2.

We now explore the penalties from resistances in the LEDs and PV cells. We evaluate the case with the AlGaN spacer layer as considered above which yields superior performance. In the presence of series resistances ($R_{s,LED}$ and $R_{s,PV}$) and shunt resistance ($R_{sh,LED}$ and $R_{sh,PV}$), the input and output voltage can be related based on the circuit diagram in Fig. 1c as

$$V_{IN} = V_{LED} + (I_{LED} + \frac{V_{LED}}{R_{sh,LED}/A_{LED}})\frac{R_{s,LED}}{A_{LED}} \qquad (10)$$

and

$$V_{OUT} = N[V_{PV} + (I_{PV} + \frac{V_{PV}}{R_{sh,PV}/A_{PV}})\frac{R_{s,PV}}{A_{PV}}] \qquad (11)$$

In general, the shunt resistance can be engineered to the extent that the resulting penalty on the efficiency is negligible. Therefore, in the following, we focus on evaluating the impact of series resistance and assume the shunt resistances are infinitely large for both LED and the PV cell. For GaN LEDs, a series resistance as small as 1 mΩ · cm² has been demonstrated experimentally[24], and GaN tunnel junctions with series resistance as low as 0.01 mΩ · cm² have been demonstrated as well[40]. In Fig. 2b, we show the modeled photonic transformer performance for series resistances of 1, 0.1, and 0.01 mΩ · cm². In general, the performance

degrades as the series resistance goes up. For high-input voltages above 3.2 V, the series resistance penalty becomes the dominant limitation on the voltage-conversion ratio and further increasing the input voltage leads to diminished performance. However, the voltage-conversion ratio at the maximum efficiency point can still reach over 90 (for $N = 100$) for all three cases, and the peak efficiency can exceed 90% when $R_s = 0.01$ mΩ · cm², indicating that the excellent performance persists even in the presence of realistic series resistance. We note that, in theory, conventional switching converters can also have a theoretically arbitrary voltage-conversion ratio by controlling the duty cycle. However, parasitic losses in the circuit[1] typically place a severe limit on the useful range of conversion ratios to the order of ten[41,42]. In contrast, the conversion ratio of our photonic transformer should not be subject to such a limit. The high conversion ratio and efficiency indicate the great potential for photonic transformers to outperform conventional switching converters[42].

Based on the above analysis, we propose a conceptual monolithic solid-state device design illustrated in Fig. 2c. Separating the GaN LED layers and PV cell layers is the index-matching AlGaN layer that provides both the necessary optical coupling and electrical insulation. Since the LEDs and PV cells can be readily miniaturized and monolithically integrated on a single die, the photonic transformer can be made with a far

smaller footprint and lower weight than those of existing switching DC converters. The design concept shown in Fig. 2c uses one LED. Alternatively, one can use multiple LEDs connected in parallel, which have the same theoretical performance, provided that the total emitting areas of the LEDs are the same. In practice, using multiple LEDs may facilitate the design of better optical coupling, and allow better control over the voltage-conversion ratio.

In addition to its high performance in terms of conversion ratio and efficiency, the use of a steady photon flux enables the photonic transformer to produce an ultralow ripple output voltage. The contributions from the photonic transformer to the output voltage fluctuations are primarily the photon shot noise and the thermal noise from the series resistance, both of which are fundamental in nature. These noises have broad frequency spectra. But even when integrated over the entire frequency bandwidth, the power in such noises is still small in comparison with typical thermal noise power associated with a typical load resistance ("Methods"). This contrasts with the switching converters where the output voltage ripple, as well as the accompanying EMI, are an unavoidable result of switching.

**Proof-of-concept demonstration of photonic transformer**. To validate the above theoretical model of the photonic transformer, we construct a circuit prototype and test its performance using commercially available off-the-shelf LEDs and PV cells. We use multiple LEDs connected in parallel, as shown in the printed circuit board (PCB) design in Fig. 4a. Here, we use the same number ($N$) of the LEDs and the PV cells. We choose $N = 100$ to show the high conversion ratio that the photonic approach enables. We use GaAs LEDs and Si PV cells to ensure reasonable spectral overlap between the LED and PV cell. We note that these choices are only for demonstration purposes and are far from the optimized devices discussed above. Figure 4b shows the LED and PV cell circuits and the assembled prototype consisting of the LED board facing the corresponding PV cell board (see "Methods" for the circuit construction).

To characterize the photonic transformer prototype, we connect the LED board to a DC power supply and measure the current–voltage ($I–V$) curve of the PV cell at different input voltage levels. We obtain the maximum efficiency of the transformer by locating the maximum power point of the PV cell array on the measured $I–V$ curve. In Fig. 4c, we show the voltage ratio at the maximum efficiency point and the corresponding efficiency of the transformer for different input voltages. The efficiency peaks at the input voltage of $V_{IN} = 1.39$ V. Further increasing the input voltage leads to a decrease in efficiency due to the series resistance of the LEDs and the PV cells, as discussed earlier in the theoretical calculation shown in Fig. 2c. At the peak efficiency (5.77%), we obtain a voltage-conversion ratio 31.2, a clear demonstration of the DC voltage-conversion functionality of our photonic transformer. The ratio between the open-circuit voltage and the input voltage is 40.9. Therefore, one could tune the operation point to obtain an even higher voltage-conversion ratio. We analyze the crude prototype photonic transformer circuit using the proposed model. The predictions are shown in Fig. 4c as continuous curves, which agree well with the experimentally measured values. This agreement provides validation of our theoretical model. As an interesting side note, our demonstrated voltage-conversion ratio (>40) exceeds what has been previously demonstrated in existing experiments on photonic transformers (<10)[10,11].

From the device model for the prototype, we identify several aspects that could be improved for better performance. These aspects include the series resistance of the PV, the external

quantum efficiency for both the LED and the PV, and the optical coupling between the LED and the PV cell. The series resistance in our device ($R_{s,PV} = 2.33\ \Omega\ cm^2$) can be improved significantly to as small as $0.32\ \Omega\ cm^2$ by optimizing the PV cell design[23]. This improvement especially helps to minimize the loss at high-power levels as shown by the blue curve in Fig. 4d. Also, one can use higher-quality semiconductor materials to improve the radiative efficiency of the LED and the PV. For the LED, both the Auger process and the SRH process are important nonradiative nonidealities since the LED is operating near its peak of quantum efficiency. For the PV cell, the SRH process is the major nonradiative recombination process because its bias at relevant operating conditions is far below its bandgap. With the relevant parameters replaced by the improved numbers reported in the literature[43,44], efficiency and the voltage-conversion ratio can be both significantly improved for all input power levels, as shown by the yellow curves in Fig. 4d. In addition, the optical coupling between the LED and the PV cell can be improved. This includes improved light extraction to air for the LED, which reduces the internal photon loss inside the devices and increasing the view factor between the LED and PV cell to suppress photon leakage to the environment. With all these improvements implemented, the performance can be raised to that indicated by the purple curves in Fig. 4d where the efficiency is significantly enhanced to ~40%. These calculations indicate the pathways to improve the performance of demonstrated photonic transformer here using existing components, and further justify the necessity of the proposed monolithic design in enhancing the optical coupling and improving light-extraction efficiency to achieve the ultimate performance of the photonic transformer.

As final remarks, we measure and validate that the additive noise output voltage noise and EMI of our prototype photonic transformer are below our measurement capability while that of a comparable switching converter is clearly observed (Supplementary Fig. 2). We note that the monolithic photonic transformer can also operate as a step-down DC transformer if the LEDs are in series and PV cells are in parallel. The monolithic photonic transformer is highly scalable and can be easily integrated on-chip[45]. The conversion ratio and/or the output voltage can be modified in real time by a switch network that reconfigures the connections among the LEDs and another switch network on the PV side such that the output voltage and current can be adjusted in discrete steps. Other high-quality semiconductors[14] may be used, depending on the application and input voltage range. Our photonic transformer also inherently provides electrical isolation between the input and output, protecting the load from destructive or hazardous electric shocks. While the initial application of the photonic transformer concept is likely in low power electronic circuits (Ws to kWs level), one may envision that this concept can be scaled up to a power level relevant for electric power network (MWs level). Our photonic transformer can also be combined with conventional switching converters, to support voltage regulation, while still providing the benefits of high efficiency, low footprint and weight, and low noise. The proposed photonic transformer highlights the significant potential for using photons as the intermediate energy carrier in power conversion applications.

## Methods
**Fluctuation–dissipation theorem**. In the theoretical description of the GaN photonic transformer, we use the fluctuation–dissipation theorem to model the radiative recombination rate since this approach is applicable for both the near- and far-field scenarios, and one can describe the photon transport process without explicitly defining the radiative recombination coefficient, escape probability, and light-extraction efficiency. Here, we consider the case where the surfaces of the LED and the $N$ PV cells are parallel, and the schematic shows a unit cell of the whole structure as Fig. 3a, b. The above-bandgap emission from a diode is due to a

fluctuational current source $\boldsymbol{j}$ that satisfies[26]

$$\langle j_k(\boldsymbol{x}',\omega)j_n^*(\boldsymbol{x}'',\omega')\rangle = \frac{4}{\pi}\omega\epsilon_0\mathrm{Im}(\epsilon_e)\Theta(\omega,T,V)\delta_{kn}\delta(\boldsymbol{x}'-\boldsymbol{x}'')\delta(\omega-\omega') \quad (12)$$

where $k$ and $n$ denote the directions of polarization, $\boldsymbol{x}'$ and $\boldsymbol{x}''$ are position vectors, $\mathrm{Im}(\epsilon_e)$ is the imaginary part of the dielectric function, $\epsilon_0$ is the vacuum permitivity, $\delta$ is the Dirac delta function. Using the formalism of fluctuational electrodynamics, the energy transfer between the LED and the PV cell can be modeled as

$$Q = Q_{\mathrm{LED}\rightarrow\mathrm{PV}}(V_{\mathrm{LED}}) - Q_{\mathrm{PV}\rightarrow\mathrm{LED}}(V_{\mathrm{PV}}) \quad (13)$$

where

$$Q_{\mathrm{LED}\rightarrow\mathrm{PV}}(V_{\mathrm{LED}}) = \frac{A_{\mathrm{LED}}f_{\mathrm{LED}\rightarrow\mathrm{PV}}}{8\pi^3}\int_{\omega_g}^{\infty}d\omega\iint\xi(\omega,k_x,k_y)\Theta(\omega,V_{\mathrm{LED}})dk_xdk_y \quad (14)$$

and

$$Q_{\mathrm{PV}\rightarrow\mathrm{LED}}(V_{\mathrm{PV}}) = \frac{A_{\mathrm{PV}}f_{\mathrm{PV}\rightarrow\mathrm{LED}}}{8\pi^3}\int_{\omega_g}^{\infty}d\omega\iint\xi(\omega,k_x,k_y)\Theta(\omega,V_{\mathrm{PV}})dk_xdk_y \quad (15)$$

The photon flux can be obtained accordingly as written in the main text.

In the far-field case with air, the energy transmission coefficient is non-negligible only for $\beta^2 = k_x^2 + k_y^2 < \omega^2/c^2$, where $c$ is the speed of light in air. In contrast, in the case with an AlGaN spacer layer, there are significant contributions from regions with $\beta^2 > \omega^2/c^2$. In Supplementary Fig. 1, we show the external quantum efficiency (EQE) of the GaN LED in both the far-field case with air, and the case with AlGaN spacer layer. In the far-field case, the GaN LED has an equivalent radiative recombination coefficient about $B = 9\times10^{-12}\,\mathrm{cm}^3/\mathrm{s}$, similar to the reported typical value[27]. In the case with AlGaN spacer layer, the light-extraction efficiency is greatly enhanced, resulting in an enhanced EQE for the LED as the figure shows. We note that, by enhancing the light-extraction efficiency with the use of the AlGaN spacer layer, one also increases the radiative recombination coefficient, since previously trapped photons that are trapped and reabsorbed by the LED (contributing to a reverse current) can now be extracted. Therefore, the use of the AlGaN spacer layer also greatly improves the current density and the power density of the photonic transformer.

## Photonic transformer with N LEDs and N PV cells.
We perform an analysis on the ideal performance of the setup with identical $N$ LEDs and $N$ PVs. For the circuit shown in Fig. 4a, the LEDs have the same current and the PV cells have the same voltage. We denote the voltage (current) of each LED and PV cell as $V_{\mathrm{LED}}$ ($I_{\mathrm{LED}}$) and $V_{\mathrm{PV}}$ ($I_{\mathrm{PV}}$), respectively. Due to the symmetry of the system, the input and output voltages are respectively $V_{\mathrm{IN}} = V_{\mathrm{LED}}$ and $V_{\mathrm{OUT}} = NV_{\mathrm{PV}}$, when series resistance is neglected. The input and output currents are, respectively, $I_{\mathrm{IN}} = NI_{\mathrm{LED}}$ and $I_{\mathrm{OUT}} = -I_{\mathrm{PV}}$. Compared to the one LED and $N$ PV cells discussed in the main text, the only difference is in the formula for $I_{\mathrm{IN}}$. In the ideal case, the total current in $N$ LEDs in parallel is equal to the current in one LED, provided that the total emitting area of the $N$ LEDs is the same as the emitting area of the one LED. Therefore, $I_{\mathrm{IN}}$ is equivalent for the $N$ LEDs case and one LED case. Thus, the photonic transformer will have the same theoretical performance for the two cases. Practically, using multiple LEDs may assist the optical coupling between the LEDs and PV cells, and help eliminate the series resistance caused by current spreading in large active area LEDs.

## Construction of the photonic transformer prototype.
We design two circuit boards and have them fabricated by PCBWay—one to populate 100 LEDs (Osram SFH4253-Z GaAs LEDs) and the other to house 100 PV cells (Osram BPW 34S-Z Si PIN photodiodes). The LEDs/PV cells are arranged in a $10\times10$ grid with 1 cm pitch in either direction on the corresponding board and routed to realize a parallel (series) connection on the LED (PV) board. Power connections for both boards are made on the reverse side of the boards. The two boards are mounted with LEDs and PV cells facing each other using alignment holes placed at each board corner through which a series of bolts, spacing washers, and nuts are used to maintain LED-to-PV alignment and ensure good optical coupling. In characterizing the photonic transformer prototype, we connect the LED board to a DC power supply (Keysight E36312A) and measure the $I$–$V$ curve of the PV cell board using a source meter (Keithley 2636B) at different input voltage levels.

## Measurement of view factor and LED EQE.
The PV cell in general has a less than 100% probability of converting an incident photon into photocurrent. We denote the external quantum efficiency of the PV cell as $\eta_{\mathrm{RES}}$ to account for the nonideal response of the PV cell. Supplementary Fig. 3 shows the external quantum efficiency of the Si PV cell and the electroluminescent emission spectrum of the GaAs LED. The external quantum efficiency of the PV cell is defined as the ratio between the output electron number flux and the input photon number flux at the short-circuit condition. The response of the PV cell is characterized by the spectrum-averaged external quantum efficiency from 725 to 925 nm $\eta_{\mathrm{RES}} = 0.897$. The average photon energy of the LED emission spectrum is 1.46 eV (847 nm). Based on its datasheet, the emitted optical power from the LED is 40 mW at $I_{\mathrm{IN}} = 70$ mA. With the averaged emitted photon energy, we compute the external quantum efficiency of the LED at this input power level and find $EQE = 39.1\%$.

To measure the photon transfer efficiency from the LED to the PV cell, we build a separate device that has only one LED and one PV cell. We then measure the ratio of the input current of a LED and the short-circuit current of the PV cell as shown in Supplementary Fig. 4, from which

$$\frac{I_{\mathrm{OUT}}}{I_{\mathrm{IN}}} = EQE\times f_{\mathrm{LED}\rightarrow\mathrm{PV}}\times\eta_{\mathrm{RES}} \quad (16)$$

At $I_{\mathrm{in}} = 70$ mA, we measure a current ratio of 0.254. Together with the averaged external quantum efficiency of the PV cell, we obtain $f_{\mathrm{LED}\rightarrow\mathrm{PV}} = 0.73$. In the device model, we assume this view factor is the same for every LED and PV cell pair in the transformer prototype. We then measure the $I$–$V$ curves of the LED array as shown in Supplementary Fig. 5a. For each input level, we measure the $I$–$V$ curve of the PV cell array as shown in Supplementary Fig. 6. Based on Eq. (16), we obtain the EQE of the LED array at different input levels as shown in Supplementary Fig. 5b from the ratio of output short-circuit current and the input current. We show the voltage and the current at the peak efficiencies in Supplementary Fig. 7.

## Device model for the photonic transformer prototype.
Since our prototype is a far-field device, we can simplify the model and highlight the important non-idealities. Instead of Eq. (6), we compute the photon flux produced by the LED as

$$F_0 = A_{\mathrm{LED}}B_{\mathrm{LED}}n_{\mathrm{LED}}p_{\mathrm{LED}}t_{\mathrm{LED}} \quad (17)$$

In the above equation, $t$ is the thickness of the active region of the diode, $A$ is the area of the active region, $n$ and $p$ are the electron and hole concentrations, respectively, and $B$ is the radiative recombination coefficient. Due to the refractive index contrast between the LED and air, many of the generated photons will be trapped in the LED and eventually absorbed parasitically by the LED such as in the contacts. Therefore, we introduce a light-extraction efficiency ($\eta_{\mathrm{EXT}}$) which describes the proportion of photons that can escape from the LED into free space. The imperfect transmission of light from the LED to the active region of the PV cell is captured by the geometric view factor $f_{\mathrm{LED}\rightarrow\mathrm{PV}}$. We lump the internal photon loss in the LED and the PV cell detection photon loss all in the ambient terms in Eqs. (3) and (4). With these parameters, the photon flux terms in Eqs. (3) and (4) can be modeled as

$$F_{\mathrm{LED}\rightarrow\mathrm{PV}} = f_{\mathrm{LED}\rightarrow\mathrm{PV}}\eta_{\mathrm{EXT}}\eta_{\mathrm{RES}}F_0 \quad (18)$$

and

$$F_{\mathrm{LED}\rightarrow\mathrm{amb}} = (1-f_{\mathrm{LED}\rightarrow\mathrm{PV}})\eta_{\mathrm{EXT}}F_0 + (1-\eta_{\mathrm{EXT}})F_0 + f_{\mathrm{LED}\rightarrow\mathrm{PV}}\eta_{\mathrm{EXT}}(1-\eta_{\mathrm{RES}})F_0 \quad (19)$$

In Eq. (19), the first term on the right-hand side is the photon loss directly to the ambient, the second term is the internal photon loss in the LED, and the third term is the photon loss in the incident photon flux that is received but not absorbed by the active region of the PV. Since the photon fluxes emitted by the Si PV cell and the ambient in general are much smaller compared to that from the emission from the GaAs LED with a bias, we neglect the other photon flux terms in Eqs. (3) and (4). The nonradiative terms are the same as Eq. (9). Substituting Eqs. (18), (19), and (9) into Eqs. (3) and (4), we obtain a model for the $I$–$V$ curves of the LED and PV cell boards. Besides the parameters that are measured (i.e., view factors and LED EQE), we obtain the remaining parameters used in the model by fitting the measured $I$–$V$ curve of the LED and the set of $I$–$V$ curves of the PV cell iteratively using the *fmincon* function provided by MATLAB. We list the obtained parameters in the tables in the Supplementary section.

## Setup for measuring output voltage fluctuations and electromagnetic field emissions.
We measure and compare the electromagnetic noise generated by the conventional switching converter and the photonic transformer using the setup shown in Supplementary Fig. 2a. The setup consists of an oscilloscope to monitor the output voltage fluctuations and a field probe connected to a spectrum analyzer to monitor the emitted electromagnetic fields. For commercial switching converter design, we use Microchip MCP1640EV (Supplementary Fig. 2b), which is an evaluation board containing the manufacturer's suggested design and board layout to implement a step-up DC-to-DC converter. Each circuit undergoes the following measurement procedures: (i) an appropriate load resistance to produce ~50 mW output power is selected and mounted on the circuit output; (ii) input DC voltage supply (Keysight E36312A adjustable DC power supply) is applied; (iii) output voltage level is measured (B&K 2709B multimeter) and output voltage waveform is taken (Lecroy WaveAce 1012 oscilloscope); and (iv) field emission spectrum is taken using magnetic field probe (Beehive Electronics BH100C) connected to a spectrum analyzer (Tek 495 P). In the final step, we maintain a 2-cm parallel gap between the field probe and the circuit board; the location of the probe where the spectrum is taken is the one at which the maximum field is registered on the spectrum analyzer as measured by the magnitude of the lowest frequency peak, if available. An extra spectrum is taken with power to the circuit under test turned off to provide the measurement of the background/instrument noise floor. Measurement parameters for the switching converter and photonic transformer circuit are, respectively, as follows. Load resistance: 220 Ω, 68 kΩ; input voltage: 1.0, 1.5 V; measured output DC voltage: 3.3, 58.2 V. Spectrum analyzer settings: 1 kHz

resolution bandwidth, auto sweep rate (see Supplementary Fig. 2 for the photo of measurement setup).

**Noise analysis for the photonic transformer circuit**. We begin by describing noise processes in a photonic transformer consisting of only one LED and one PV cell (Supplementary Fig. 8a) and subsequently consider the entire system, including multiple PV cells and the load. The photon statistics of an LED connected to a voltage source is well-characterized by photon shot noise[46,47]. The noise in the PV cell at low injection levels can be considered as a result of independent noise fluctuations, each modeled as a Poissonian noise process, as shown in Supplementary Fig. 8b where $I_{\text{photo}}$ is the photocurrent, $I = I_0 e^{\frac{qV}{\eta kT}}$ is the junction's forward current, and $I_0$ is the saturation current[46,48]. Hence the combined noise is $S_{I_{\text{PV}}}(f) = 2q(I_{\text{photo}} + I + I_0) \leq 4qI_{\text{photo}}$, where the upper bound is reached at the open-circuited operation ($I_{\text{OUT}} = 0$). $R_j = \frac{\eta kT/q}{I}$ and $C_j$ are small-signal junction resistance and capacitance with $R_j C_j \approx$ minority carrier lifetime, $\tau$[49,50]. In addition, the series resistance $R_s$ and shunt resistance $R_{sh}$ contribute Johnson noise with $S_{I_{Rs}} = 4kT/R_s$ and $S_{I_{Rsh}} = 4kT/R_{sh}$. For a photonic transformer with $N$ PV cells, fluctuations from all the PV cells combine to produce noise at the load $S_V(f) = S_{V,\text{PV}}(f) + S_{V,Rs}(f) + S_{V,Rsh}(f)$, where $S_{V,\text{PV}}(f) = NS_{I_{\text{PV}}}(f)|H_{\text{PV}}(f)|^2$, $S_{V,Rs} = NS_{I_{Rs}}(f)|H_{Rs}(f)|^2$, and $S_{V,Rsh}(f) = NS_{I_{Rsh}}(f)|H_{\text{PV}}(f)|^2$ are the contributions from $S_{I_{\text{PV}}}$, $S_{I_{Rs}}$ and $S_{I_{Rsh}}$, respectively, with $H_{\text{PV}}(f) = (\frac{Z_j}{NZ_j + NR_s + R_L})R_L$ and $H_{Rs}(f) = (\frac{R_s}{NZ_j + NR_s + R_L})R_L$ being the transfer functions from their respective individual noise sources to voltage noise on the load, and $Z_j = R_{sh} || R_j || (\frac{1}{2\pi i f C_j})$ (Supplementary Fig. 8c). We evaluate these noise contributions for the photonic transformer circuit under the measurement conditions of Supplementary Fig. 2 with relevant circuit parameters as follow: $N = 100$, $I_{\text{photo}} = 9$ mA, $R_j = 6\,\Omega$, $\tau = 0.551\,\mu s$, $R_s = 31\,\Omega$, $R_{sh} = 21\,k\Omega$, and $R_L = 68\,k\Omega$. We have $S_{V,\text{PV}}(f) = 20$ nV$^2$/Hz and $S_{V,Rsh} = 0.003$ nV$^2$/Hz, both flat noise power up to $f_c = \frac{1}{2\pi\tau} = 300$ kHz, and $S_{V,Rs} = 50$ nV$^2$/Hz up to the bandwidth of the oscilloscope. These noise contributions from the photonic transformer are miniscule in comparison with a noise level from an ideal resistive load of $R_L$ which produces noise power $4kTR_L = 1100$ nV$^2$/Hz at room temperature—this statement applies in general for $R_L \gg NR_s$ which corresponds to near-constant output voltage operation. Finally, we note that low-frequency noise (often referred to as "1/f" or flicker noise), which typically shows up in electronics and manifests as fluctuations over a long time scale, may contribute to higher noise at low frequency. Such noise has been found to correlate with defects in semiconductor lattice and contacts, and can be reduced with higher quality device preparation[51,52].

## Data availability

The measured $I$–$V$ data of the photonic transformer prototype generated in this study have been deposited in the Figshare repository [https://doi.org/10.6084/m9.figshare.14729235].

## Code availability

The code used in this work is available at https://github.com/fancompute/MESH.

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

## Acknowledgements

This work was supported by the U.S. Department of Energy Photonics at Thermodynamic Limits Energy Frontier Research Center under Grant DE-SC0019140 (theoretical model), by a U.S. Army Research Office MURI project under Grant W911NF-19-1-0279 (fluctuational electrodynamic calculation), and by a U.S. Department of Defense Vannevar Bush Faculty Fellowship under Grant N00014-17-1-3030 (experiment). B. Zhao acknowledges Dr. David Miller and Dr. Avik Dutt for stimulating discussions.

## Author contributions

B.Z. and S.A. performed the simulation, modeling, and experiment. P.S. assisted with the simulations and modeling. All the authors contributed to formulating the analytical model, to analyzing the data, and to writing the manuscript. B.Z. and S.F. initiated the project. S.F. supervised the project.

## Competing interests

The authors declare no competing interests.
