## [Peer Review File · Nature Communications]

REVIEWER COMMENTS

Reviewer #1 (Remarks to the Author):

The authors present an argument for the development of the 'Photonic Transformer,' an arrangement of closely coupled light-emitting diodes (LEDs) and photovoltaic (PV) solar cells, offering a pathway to highly efficient steady-state DC voltage/power conversion, as an alternative to conventional switched-mode power converters. Compared to the prior cited art, a key difference in the author's proposed architecture seems to be that the photoreceivers are arranged laterally rather than as vertically stacked tandem devices, potentially offering greater flexibility in device design.

The notion that LEDs can be closely coupled to bandgap-matched PV cells for the delivery of power across an air gap (for power transmission or for input-output isolation) or for voltage step-up/step-down via series/parallel arrangements, is not a new one. But the authors thesis is the argument that, in principle, photonic transformers can approach unity conversion efficiency. This is somewhat remarkable, because as far as I know, the far-field DC-DC power conversion efficiency record for solar cells and lasers/LEDs is at most 60%.

The authors' theoretical analysis should be of good interest to the scientific community. If realized, high-efficiency photonic transformers could revolutionize many types of power conversion circuits, especially for low power, miniature, or lightweight applications. In particular, the authors' vision of an integrated photonic transformer device (Fig. 1g) represents in my opinion a promising frontier for the future of power converters.

However, I am less than enthusiastic about the presentation of experimental work, to the degree that I fear it detracts somewhat from the message and impact of the paper. In my opinion, the discussion of the proof-of-concept device (Fig 2) focuses too much on the fanfare of its existence, and misses a crucial opportunity to deliver an even more important message: Not only does the prototype's performance agree with the authors' efficiency prediction, but their model also offers a great deal of insight into why the present-day efficiency is so low, and thus hints at how it can be improved (!). Why is this fact buried in the methods and SI?

I'm specifically talking about Figure S7, which shows how the three key compromises arising from the off-the-shelf parts (bandgap mismatch, optical losses, and resistance losses) reduced its efficiency. This suggests that straightforward engineering efforts could produce a device with efficiency exceeding 40%. So, although the authors can rightfully proclaim that their choice of 100 coupled diodes achieved the best-ever voltage ratio compared to prior devices made with fewer numbers of differently arranged diodes, I feel like this is a distraction from the more impressive fact that they've presented a highly scalable device architecture capable of even better ratios and efficiencies, and have contributed theoretical and practical insights that can help guide future research towards this goal. Figure S7 and its supporting discussion absolutely belong here in the main manuscript body!

Unfortunately, I believe that the treatment of the experiment in Figure 3 should either be substantially expanded to provide better context, or omitted from the paper. There are two reasons for this.

First, although I appreciate that the authors have performed careful and thorough measurements here, I don't think the results are sufficiently novel or noteworthy to be featured in a Nat Comm paper. The fact that switching-mode power converters emit switching noise/EMI, whereas non-switching linear devices don't, is common knowledge and has also been shown in prior work (e.g., ref 10). If the paper were describing a new exciting kind of electric car, would there be a pressing need to carefully measure and compare its tailpipe emissions to those of two gas-powered cars? I would argue no, because it is commonly known that electric cars don't have tailpipes. So although I don't mean to criticize the quality and detail of work that clearly went into this particular set of experiments, what I ultimately see in Figure 3 are three virtually identical noise spectra, with the exception of course that the switching devices also exhibit switching EMI. And this is

nowhere near as novel or exciting as the rest of the paper.

But here's the bigger reason I don't like the discussion surrounding Figure 3: In comparing the photonic transformer to commercial switched-mode power converters, the authors overlook a key difference between these two types of circuits. In particular, the photonic transformer has no mechanism to regulate or vary its output vs. input voltage in response to changing conditions, whereas the switch-mode converters operate over a wide range of input and output voltages, and over a wide range of load currents, providing precise output voltage regulation that varies by only a few 10s of mV over the entire industrial temperature range.

It is tempting to write off this shortcoming of the photonic transformer as a result of its technological infancy, or as a result of limited early-stage prototyping. But the underlying fact is that switch-mode power converters can vary the frequency and duty cycle of the switching process to efficiently convert power over a wide range of input and output voltages or currents, whereas the output of the photonic transformer is essentially determined by its input excitation and its physical configuration, i.e., the number of PV cells or LEDs used in its construction. A second-order yet noteworthy issue is that the output voltage of the PV cells (or the input current of the LEDs) will vary considerably with device temperature, whereas switching converters can easily (and routinely do) incorporate precise temperature-compensated references in their regulation circuitry. Ultimately, in the high-efficiency limit, the photonic transformer will necessarily become a fully reciprocal device (just like the inductive transformer is for time-varying signals), which is quite interesting, but neither type of transformer comprises a properly regulated power converter on its own.

So while in principle the photonic transformer is indeed a non-switching "EMI free" DC-DC converter device, it seems unfair to extol this virtue in such great experimental detail without also addressing the requirements of line, load, and thermal regulation in practical power conversion applications. I thus suggest that the authors either omit the detailed experimental EMI measurements in the present paper, or greatly expand it to more thoroughly address the holistic design of regulated photonic power converters, considering not only EMI but also line and load regulation and its effects on end-to-end efficiency over wide ranges of operating conditions.

For example, for high voltage converters with high voltage ratios, it might make sense to integrate bypass switches in the LEDs or the PV cells, so that the specific ratio of LEDs to PV cells could be dynamically varied in response to changes in the load or supply voltage. This 'variable turns ratio' approach could be used to provide some input and output voltage regulation, but only in steps on the order of the semiconductor bandgap, yet would allow for pseudo-steady-state DC-mode power conversion without additional external regulators or converters—at least assuming that a reasonably efficient light distribution architecture could be devised. Given realistic values for radiative and light-coupling efficiencies, what would the regulation and efficiency vs. load curves look like for such a device, vs. that of typical switch-mode converters?

Alternately, if a finer degree of voltage regulation were required, could the photonic transformer itself be operated in a switched mode (e.g., PWM) with the addition of a regulated driver circuit and power-smoothing elements? Would the large junction capacitance and forward charge injection in both diodes lead to unfavorable switching losses? Alternately, could the photonic transformer simply be connected to a conventional switch-mode power converter, while still providing some degree of benefit in terms of efficiency, footprint, EMI, etc.?

I think would be quite reasonable to retain the EMI measurements as SI if the authors wish, but I ultimately believe that any detailed discussion of EMI comparisons within the main manuscript body should be accompanied by correspondingly detailed treatment of at least some of the issues I've outlined here.

Reviewer #2 (Remarks to the Author):

The authors present work on a number of items regarding photonic power converters. The paper starts with a detailed balance analysis which shows that for a monolithically integrated design operating with a near-

field mode, the converter could operate at the radiative limit. In this case, the near-field spacer material would be AlGaIn. The paper then goes on to show a proof of concept using GaN LEDs and Si photovoltaics. In the main text, it is difficult to figure out what component was used as one must go to Extended Data Table 1 to find this information. The experimental demonstration is different from the concept explained earlier in the paper. The proof of concept uses an array of LEDs and Si detectors both mounted on separate bread boards and the interface between them is an air gap. The detectors are connected in series in order to demonstrate an output voltage of 33 V. The demo dimensions are approximately 15 cm x 20 cm x 1 cm and cannot be considered an integrated optoelectronic chip. Figure 3 demonstrates output from a commercial switching converter and of the proof of concept photonic converter. Results obtained are similar than those in reference 10, which is made with an integrated series-connected GaAs photovoltaics instead of breadboard assembled series-connected Si photovoltaics. The novelty here is that it is demonstrated with GaN based LEDs and Si photovoltaics. Because so many Si cells are connected in series, one obtains a higher voltage, 33 V as opposed to 12 V reported in reference 10. The wall plug efficiency reported here is 5.8% while in reference 11 it was 27%.

With regards to the title, the authors talk about photonic transformers and then in the paper, they explain why they think it's a transformer. When one looks at the definition of a transformer versus that of a converter, and what a converter is used for, in my opinion, this device is a converter. This should be changed throughout the paper.

The work is sound and the paper is of interest to the community but is difficult to follow the way it's written. In order for it to be of interest to a wider community, the reviewer would recommend focusing on only one of the two concepts, either the theory for a GaN based integrated system or the experimental validation of a different system of the GaN LED – Si PV. For the theory, the authors should refer to this early paper by Martin Green, which gives more generic similar assertions than the results presented in figure 1: M. A. Green, "Limiting photovoltaic monochromatic light conversion efficiency", *Progress Photovolt.: Res. Appl.*, vol. 9, no. 4, pp. 257-261, 2001.

Reviewer #3 (Remarks to the Author):

The manuscript by Zhao, et al. reports a photonic DC voltage converter using LEDs and PV cells. Based on detailed balance analysis, this photonic transformer can operate with high conversion efficiency and voltage ratio. The authors validate the theoretical analysis of the proposed transformer with experimental demonstration using off-shelf components.

The manuscript is well-written with appropriate background and motivation. The proposed photonic transformer is an innovative and promising optical solution to miniaturizing DC power converter. I have several questions and comments about various claims and methodologies in the manuscript that should be addressed before publication:

1. The authors claim in the manuscript that the GaN PV cell can recover 97% of the voltage of the GaN LED with a power conversion efficiency of over 85% as shown in Fig. 1d. based on the theoretical analysis provided in Method. However, I could not find relevant content in Method that explicitly supports this claim. It seems the most relevant part that I should look at is the "Fluctuation-dissipation theorem". Through Eqs. (M1) to (M2), I get that the emission of the LED and the absorption of the PV are driven by the term $\Theta(\omega, T, V)$; Eqs. (M3) to (M5) describe the energy transfer between LED and PV cell; Eqs. (M6) to (M8) describe the balance of photon flux. My problem is, how are these equations explicitly point to the relation between V_{PV} and V_{LED} ? Are there any additional information (such as GaN optical properties, emission spectrum of the LED, or the absorption spectrum of the PV cell) that are required to obtain Fig. 1d. (middle)? In other word, maybe the authors can provide more explanation (mathematically or verbally) on how to combine these Eqs. to solve for the relation between V_{LED} and V_{PV} and generate Fig. 1d (middle).
2. It is also confusing how Fig. 1d (right) is plotted. The authors state that "Using Eqs. (M6), (M7), and

(M8), together with Eqs. (3) and (4), we obtain a model of the I-V curve for both the LED and PV cell". The author should provide some intermediate steps or figures to explain how these I-V curves are obtained. For example, the emission spectrum of the LED and the absorption spectrum of the PV, the current integrated from their spectral overlap, and plots of the calculated I-V curves for both the LED and PV cell would be very helpful.

3. In Extended Data Table 1, the authors fit the SRH lifetime of GaAs LED and Si PV to be 22 ns and 92.8 ns, respectively. These values imply extremely poor materials quality, and will lead to considerable degradation in turn-on voltage of the LED and Voc of the PV. However, from the Extended Data Figure 4 and 5, the I-V curves of the commercial LED and PV cells look normal (good turn-on voltage for the LED and good Voc for the PV). I suspect that this is due to the authors ignoring the shunt resistance in these devices, and instead, using SRH current as the dominating non-ideal current source to fit the I-V curves that deviate from ideal diodes. I encourage the authors to fit the curves including the shunt resistance, in which case, the SRH lifetime should be orders of magnitude higher in these devices.

Authors' response for Zhao et al., manuscript ID NCOMMS-20-45671A. Detailed response to reviewers' comments. Reviewers' comments are in black and the responses are in blue.

Summary of list of changes in the main text:

(1) We modified two sentences in the paragraph on Page 2, and they read

“Similar to a magnetic transformer, this technique naturally provides electric isolation between the input and output ports. Therefore, in this paper, we refer to such a photon-based voltage converter as a photonic transformer. While the electromagnetic field of the photons are time-varying, the frequencies of such variations are several orders of magnitude larger than what one can detect using electronic circuits. As a result, a photonic transformer operates as a DC device.”

“The fact that the limiting efficiency of PV cells for converting near monochromatic light to electricity can approach unity [12], as well as the high theoretical efficiency of LEDs, make this technology attractive.”

(2) We added several new paragraphs discussing the details of the theoretical model from Page 6-8. Please refer to the highlighted part in the revised manuscript.

(3) We added several sentences on Page 9. They read

“As an illustration, we show the energy transmission coefficients for the far-field case with air and the case with AlGa_N spacer layer in the Figs. 3c and 3d, respectively. We see that in the far-field case with air, only the channels above the light line of air (green dashed line) contribute to photon transport. In contrast, for the case with AlGa_N layer, the channels between the light lines of Ga_N and air significantly contribute to the transport between the LED and the PV cell. The additional contribution of these channels greatly enhances the light extraction efficiency of the LED such that the external quantum efficiency of the LED improves to 98.7%, as compared with that of 90.3% for the far-field case with an air spacer (Extended Data Figure 1).”

(4) We added several sentences and modified Eqs.(10) and (11). They read

“ In the presence of series resistances ($R_{s,LED}$ and $R_{s,PV}$) and shunt resistances ($R_{sh,LED}$ and $R_{sh,PV}$), the input and output voltage can be related based on the circuit diagram in Fig. 1c as

$$V_{IN} = V_{LED} + \left(I_{LED} + \frac{V_{LED}}{R_{sh,LED}/A_{LED}} \right) \frac{R_{s,LED}}{A_{LED}} \quad (1)$$

and

$$V_{OUT} = N \left[V_{PV} + \left(I_{PV} + \frac{V_{PV}}{R_{sh,PV}/A_{PV}} \right) \frac{R_{s,PV}}{A_{PV}} \right] \quad (2)$$

In general, the shunt resistance can be engineered to the extent that the resulting penalty on the efficiency is negligible.”

(5) We added one new paragraph starting from Page 13. Please refer to the highlighted part in the revised manuscript.

(6) We added one new paragraph starting from Page 14. Please refer to the highlighted part in the revised manuscript.

(7) We added one sentence in the ending paragraph. It reads

“Our photonic transformer can also be combined with conventional switch-mode converters, to support voltage regulation, while still providing the benefits of high efficiency, low footprint and weight, and low noise.”

(8) We modified the noise analysis on Page 26. Please refer to the highlighted part in the revised manuscript.

(9) We modified Extended Data Fig. 8 and Table 1 on Pages 39-40. Please refer to the highlighted part in the revised manuscript.

(10) We modified Figs. 1, 2, 3, and 4 and their captions. Please refer to the highlighted part in the revised manuscript.

Reviewer #1 (Remarks to the Author):

The authors present an argument for the development of the ‘Photonic Transformer,’ an arrangement of closely coupled light-emitting diodes (LEDs) and photovoltaic (PV) solar cells, offering a pathway to highly efficient steady-state DC voltage/power conversion, as an alternative to conventional switched-mode power converters. Compared to the prior cited art, a key difference in the author’s proposed architecture seems to be that the photoreceivers are arranged laterally rather than as vertically stacked tandem devices, potentially offering greater flexibility in device design.

The notion that LEDs can be closely coupled to bandgap-matched PV cells for the delivery of power across an air gap (for power transmission or for input-output isolation) or for voltage step-up/step-down via series/parallel arrangements, is not a new one. But the authors thesis is the argument that, in principle, photonic transformers can approach unity conversion efficiency. This is somewhat remarkable, because as far as I know, the far-field DC-DC power conversion efficiency record for solar cells and lasers/LEDs is at most 60%.

The authors’ theoretical analysis should be of good interest to the scientific community. If realized, high-efficiency photonic transformers could revolutionize many types of power conversion circuits, especially for low power, miniature, or lightweight applications. In particular, the authors’ vision of an integrated photonic transformer device (Fig. 1g) represents in my opinion a promising frontier for the future of power converters.

>> We thank the reviewer for the support of our work and the very insightful comments. In response to the reviewer’s comment we have rewritten significant parts of the paper. Below we will address the reviewer’s suggestions point by point.

However, I am less than enthusiastic about the presentation of experimental work, to the degree that I fear it detracts somewhat from the message and impact of the paper. In my opinion, the discussion of the proof-of-concept device (Fig 2) focuses too much on the fanfare of its existence, and misses a crucial opportunity to deliver an even more important message: Not only does the prototype’s performance agree with the authors’ efficiency prediction, but their model also offers a great deal of insight into why the present-day efficiency is so low, and thus hints at how it can be improved (!). Why is this fact buried in the methods and SI?

I’m specifically talking about Figure S7, which shows how the three key compromises arising from the off-the-shelf parts (bandgap mismatch, optical losses, and resistance losses) reduced its efficiency. This suggests that straightforward engineering efforts could produce a device with efficiency exceeding 40%. So, although the authors can rightfully proclaim that their choice of 100 coupled diodes achieved the best-ever voltage ratio compared to prior devices made with fewer numbers of differently arranged diodes, I feel like this is a distraction from the more impressive fact that they’ve presented a highly scalable device architecture capable of even better ratios and efficiencies, and have contributed theoretical and practical insights that can help

guide future research towards this goal. Figure S7 and its supporting discussion absolutely belong here in the main manuscript body!

>> We thank the review for the suggestions of reorganizing the theoretical and experimental aspects of the paper. We have moved (originally) Fig. S7 to the main text as Fig. 4d. We have also added the discussion on how our experimental prototype can be improved into the main text. Please refer to items 5 and 6 listed above in the list of changes.

Unfortunately, I believe that the treatment of the experiment in Figure 3 should either be substantially expanded to provide better context, or omitted from the paper. There are two reasons for this.

First, although I appreciate that the authors have performed careful and thorough measurements here, I don't think the results are sufficiently novel or noteworthy to be featured in a Nat Comm paper. The fact that switching-mode power converters emit switching noise/EMI, whereas non-switching linear devices don't, is common knowledge and has also been shown in prior work (e.g., ref 10). If the paper were describing a new exciting kind of electric car, would there be a pressing need to carefully measure and compare its tailpipe emissions to those of two gas-powered cars? I would argue no, because it is commonly known that electric cars don't have tailpipes. So although I don't mean to criticize the quality and detail of work that clearly went into this particular set of experiments, what I ultimately see in Figure 3 are three virtually identical noise spectra, with the exception of course that the switching devices also exhibit switching EMI. And this is nowhere near as novel or exciting as the rest of the paper.

But here's the bigger reason I don't like the discussion surrounding Figure 3: In comparing the photonic transformer to commercial switched-mode power converters, the authors overlook a key difference between these two types of circuits. In particular, the photonic transformer has no mechanism to regulate or vary its output vs. input voltage in response to changing conditions, whereas the switch-mode converters operate over a wide range of input and output voltages, and over a wide range of load currents, providing precise output voltage regulation that varies by only a few 10s of mV over the entire industrial temperature range.

It is tempting to write off this shortcoming of the photonic transformer as a result of its technological infancy, or as a result of limited early-stage prototyping. But the underlying fact is that switch-mode power converters can vary the frequency and duty cycle of the switching process to efficiently convert power over a wide range of input and output voltages or currents, whereas the output of the photonic transformer is essentially determined by its input excitation and its physical configuration, i.e., the number of PV cells or LEDs used in its construction. A second-order yet noteworthy issue is that the output voltage of the PV cells (or the input current of the LEDs) will vary considerably with device temperature, whereas switching converters can easily (and routinely do) incorporate precise temperature-compensated references in their regulation circuitry. Ultimately, in the high-efficiency limit, the photonic transformer will necessarily become a fully reciprocal device (just like the inductive transformer is for time-

varying signals), which is quite interesting, but neither type of transformer comprises a properly regulated power converter on its own.

So while in principle the photonic transformer is indeed a non-switching “EMI free” DC-DC converter device, it seems unfair to extol this virtue in such great experimental detail without also addressing the requirements of line, load, and thermal regulation in practical power conversion applications. I thus suggest that the authors either omit the detailed experimental EMI measurements in the present paper, or greatly expand it to more thoroughly address the holistic design of regulated photonic power converters, considering not only EMI but also line and load regulation and its effects on end-to-end efficiency over wide ranges of operating conditions.

For example, for high voltage converters with high voltage ratios, it might make sense to integrate bypass switches in the LEDs or the PV cells, so that the specific ratio of LEDs to PV cells could be dynamically varied in response to changes in the load or supply voltage. This 'variable turns ratio' approach could be used to provide some input and output voltage regulation, but only in steps on the order of the semiconductor bandgap, yet would allow for pseudo-steady-state DC-mode power conversion without additional external regulators or converters—at least assuming that a reasonably efficient light distribution architecture could be devised. Given realistic values for radiative and light-coupling efficiencies, what would the regulation and efficiency vs. load curves look like for such a device, vs. that of typical switch-mode converters?

Alternately, if a finer degree of voltage regulation were required, could the photonic transformer itself be operated in a switched mode (e.g., PWM) with the addition of a regulated driver circuit and power-smoothing elements? Would the large junction capacitance and forward charge injection in both diodes lead to unfavorable switching losses? Alternately, could the photonic transformer simply be connected to a conventional switch-mode power converter, while still providing some degree of benefit in terms of efficiency, footprint, EMI, etc.?

I think would be quite reasonable to retain the EMI measurements as SI if the authors wish, but I ultimately believe that any detailed discussion of EMI comparisons within the main manuscript body should be accompanied by correspondingly detailed treatment of at least some of the issues I've outlined here.

>> In response to the comments, we have now moved the EMI results, originally in Fig. 3 of the main text to the supplement (Extended Data Figure 7), and further simplified the discussions by focusing on only one commercial switch-mode converter. We have also moved Fig. S1 in the original version, which is about the physical mechanism of the enhanced performance proposed in the monolithic design, to the main text as Figure 3, so that we can focus on the theoretical advances proposed in this paper.

In our original manuscript, the comparison between the commercial switched mode converter and our basic scheme of the photonic transformer is incomplete, as the reviewer rightfully commented. Compare with the switch mode converter, which is a mature technology incorporating many important functionalities such as voltage regulations and temperature stabilizations, the development of our photonic transformer is at a far more preliminary stage.

Thus, a comprehensive comparison is difficult. Instead, we only provide a few aspects of such comparisons, when appropriate, such as EMI or footprint, in various part of the main text.

We also want to acknowledge the comments by the reviewer on potentially implementing bypass switches or PWM techniques, that may be of great practical importance on the subsystem level. We have added some discussion in the concluding remarks of the main text on the potential of using photonic transformers together with traditional converters while still having benefits of low footprint, weight, and high efficiency. Please refer to items 2, 3, 5, 6, 8 listed above in the list of changes.

Reviewer #2 (Remarks to the Author):

The authors present work on a number of items regarding photonic power converters. The paper starts with a detailed balance analysis which shows that for a monolithically integrated design operating with a near-field mode, the converter could operate at the radiative limit. In this case, the near-field spacer material would be AlGa_N. The paper then goes on to show a proof of concept using GaN LEDs and Si photovoltaics. In the main text, it is difficult to figure out what component was used as one must go to Extended Data Table 1 to find this information. The experimental demonstration is different from the concept explained earlier in the paper. The proof of concept uses an array of LEDs and Si detectors both mounted on separate bread boards and the interface between them is an air gap. The detectors are connected in series in order to demonstrate an output voltage of 33 V. The demo dimensions are approximately 15 cm x 20 cm x 1 cm and cannot be considered an integrated optoelectronic chip. Figure 3 demonstrates output from a commercial switching converter and of the proof of concept photonic converter. Results obtained are similar than those in reference 10, which is made with an integrated series-connected GaAs photovoltaics instead of breadboard assembled series-connected Si photovoltaics. The novelty here is that it is demonstrated with GaN based LEDs and Si photovoltaics. Because so many Si cells are connected in series, one obtains a higher voltage, 33 V as opposed to 12 V reported in reference 10. The wall plug efficiency reported here is 5.8% while in reference 11 it was 27%.

>> We thank the reviewer for the suggestions and concerns on the structure of our paper. In response to the reviewer's comment, we have significant rewritten the manuscript by primarily focusing on the theoretical model and the novel design based on GaN technology. The experiments are presented mainly as a validation of the theoretical model. And we have significantly reduced the discussions on the experimental parts.

With regards to the title, the authors talk about photonic transformers and then in the paper, they explain why they think it's a transformer. When one looks at the definition of a transformer versus that of a converter, and what a converter is used for, in my opinion, this device is a converter. This should be changed throughout the paper.

>> Similar to a magnetic transformer, the photonic technique naturally provides electric isolation between the input and output ports. Therefore, we respectfully request that we keep the term “photonic transformer” in the manuscript. Please refer to item 1 listed above in the list of changes.

The work is sound and the paper is of interest to the community but is difficult to follow the way it's written. In order for it to be of interest to a wider community, the reviewer would recommend focusing on only one of the two concepts, either the theory for a GaN based integrated system or the experimental validation of a different system of the GaN LED – Si PV. For the theory, the authors should refer to this early paper by Martin Green, which gives more generic similar assertions than the results presented in figure 1: M. A. Green, "Limiting photovoltaic monochromatic light conversion efficiency", Progress Photovolt.: Res. Appl., vol. 9, no. 4, pp. 257-261, 2001.

>> We have followed the reviewer's suggestion and reorganize the paper to mainly focus on the GaN based integrated system. We have moved Fig. S1 to the main text as the new Fig. 3 to better illustrate the physical mechanism of the enhanced performance compared to the far-field air gap case. The mentioned paper echoes our argument of the ultimate high performance the photonic transformer can achieve, and we have added it as Ref. 12 in the revised paper. Please refer to items 1,2,3,5,6 listed above in the list of changes.

Reviewer #3 (Remarks to the Author):

The manuscript by Zhao, et al. reports a photonic DC voltage converter using LEDs and PV cells. Based on detailed balance analysis, this photonic transformer can operate with high conversion efficiency and voltage ratio. The authors validate the theoretical analysis of the proposed transformer with experimental demonstration using off-shelf components.

The manuscript is well-written with appropriate background and motivation. The proposed photonic transformer is an innovative and promising optical solution to miniaturizing DC power converter. I have several questions and comments about various claims and methodologies in the manuscript that should be addressed before publication:

>> We thank the reviewer for the strong support of our work and the comments. Below we will address the reviewer's suggestions point by point.

1. The authors claim in the manuscript that the GaN PV cell can recover 97% of the voltage of the GaN LED with a power conversion efficiency of over 85% as shown in Fig. 1d. based on the theoretical analysis provided in Method. However, I could not find relevant content in Method that explicitly supports this claim. It seems the most relevant part that I should look at is the “Fluctuation-dissipation theorem”. Through Eqs. (M1) to (M2), I get that the emission of the LED and the absorption of the PV are driven by the term $\Theta(\omega, T, V)$; Eqs. (M3) to (M5)

describe the energy transfer between LED and PV cell; Eqs. (M6) to (M8) describe the balance of photon flux. My problem is, how are these equations explicitly point to the relation between V_{PV} and V_{LED} ? Are there any additional information (such as GaN optical properties, emission spectrum of the LED, or the absorption spectrum of the PV cell) that are required to obtain Fig. 1d. (middle)? In other word, maybe the authors can provide more explanation (mathematically or verbally) on how to combine these Eqs. to solve for the relation between V_{LED} and V_{PV} and generate Fig. 1d (middle).

>> Per the reviewer's suggestion, we now have moved part of the "Fluctuation-dissipation theorem" section that is related to the model of the IV curve into the main text.

Based on the fluctuation-dissipation theorem, we can compute the photon exchange without separately computing the emission spectrum of the LED and the absorption spectrum of the PV cell. In the monolithic design we proposed, the emission from the LED and the absorption in the PV cells are strongly coupled processes due to the multiple scattering of photons between the LED's and PV cells, and one could not compute the emission of the LED and absorption of the PV cell separately then combine them to achieve an accurate calculation. As mentioned in the main text, the energy transmission coefficient contains the information of the optical coupling between the LED and the PV cell, and it is computed using the optical properties of GaN and AlGaIn taken from Ref. 24.

Using Eqs.(6), (7), and (9), one can compute the photon flux terms and the nonradiative terms. Plugging in these terms in Eqs. (3) and (4), one obtains an equation between V_{LED} , V_{PV} , and I_{LED} , and the other equation between V_{LED} , V_{PV} , and I_{PV} . In computing the solutions, one starts by fixing V_{LED} to a specific value, and then solves I_{LED} and I_{PV} at different V_{PV} . In this way, one can obtain the input power from the LED and the output power from the PV cell when the load on the PV cell is changing. The power conversion efficiency then can be calculated by the ratio of the output and input power. Once we know the maximum efficiency point, one can find the V_{PV} at this point, and thus the voltage conversion ratio at this point can be obtained accordingly. By repeating this process for different V_{LED} , one obtains the curve shown in Fig. 2a (left). We have added related discussions about the process of generating the curves in the main text, and point out that more detailed discussions on the computational aspects of fluctuation-dissipation theorem can be found in (Ref. 28) that we already cited. Please refer to item 2 listed above in the list of changes.

2. It is also confusing how Fig. 1d (right) is plotted. The authors state that "Using Eqs. (M6), (M7), and (M8), together with Eqs. (3) and (4), we obtain a model of the I-V curve for both the LED and PV cell". The author should provide some intermediate steps or figures to explain how these I-V curves are obtained. For example, the emission spectrum of the LED and the absorption spectrum of the PV, the current integrated from their spectral overlap, and plots of the calculated I-V curves for both the LED and PV cell would be very helpful.

>> As discussed in the last comment, one can fix V_{LED} first and search for the maximum efficiency point based on the detailed balance relation in Eqs. (3) and (4). By repeating the

searching process for different V_{LED} , one obtains the curve shown in the new Fig. 2a (right) (previously Fig.1d (right)). The optical coupling is lumped into the energy transmission coefficient, which is obtained from solving the Green's function of the whole system, as detailed in (Ref. 28) that we already cited. We have added more discussions about the process of generating the curves in the main text. Please refer to item 2 listed above in the list of changes.

3. In Extended Data Table 1, the authors fit the SRH lifetime of GaAs LED and Si PV to be 22 ns and 92.8 ns, respectively. These values imply extremely poor materials quality, and will lead to considerable degradation in turn-on voltage of the LED and Voc of the PV. However, from the Extended Data Figure 4 and 5, the I-V curves of the commercial LED and PV cells look normal (good turn-on voltage for the LED and good Voc for the PV). I suspect that this is due to the authors ignoring the shunt resistance in these devices, and instead, using SRH current as the dominating non-ideal current source to fit the I-V curves that deviate from ideal diodes. I encourage the authors to fit the curves including the shunt resistance, in which case, the SRH lifetime should be orders of magnitude higher in these devices.

>> Per the reviewer's suggestion, we have updated our device model to include the shunt resistances. In doing so, the SRH lifetime indeed improved significantly, especially for the PV cells. We have updated the related text, plots, and Extended Data Table 1 with the new obtained parameters. We note that even with the shunt resistances, the series resistance is still the major contributor to the loss in the device. We have also added the related discussion in the main text. Please refer to items 4 and 9 listed above in the list of changes.

REVIEWERS' COMMENTS

Reviewer #1 (Remarks to the Author):

In my opinion, the authors have substantially improved the quality of the manuscript in revision, and the result seems well suited for publication in Nature Communications, subject to minor revisions as noted below.

I would suggest minor revisions to the closing paragraph on pages 14/15. In particular, I think the authors should clarify a bit more what is meant by "electric power network" (L335)... is that intended to mean power electronics currently addressed by switching power converters, e.g., consumer power supplies, chargers, etc. (say, Ws to kW), or is this meant to imply transmission and distribution applications within the power grid (MWs?) While there are interesting questions as to the technoeconomic viability of photonic transformer at increasing scale, I don't think that needs to be addressed here. My comment is motivated simply by the fact that "electric power network" seems to have a really vague meaning.

Also, I have never heard the term "fabric" used to describe a circuit topology in this context (L329). Is this intended to describe a specific switching configuration, or an additional material/layer between the solar cells and the LEDs that could otherwise change the behavior? Ultimately I feel that a few more sentences should be added to describe the capabilities afforded by the proposed approaches, in terms of voltage/current adjustment range (e.g., discrete, rectangular ratio steps, or continuous variability). Without further conceptual development, it's not clear to me that electrical switching networks alone would be capable of allowing arbitrary LED-to-PV ratios without impacting the efficiency of the device. I'm not claiming that the authors need to present any more conceptual developments in the present manuscript, just more clearly state the capabilities and limitations of the proposed approaches, and the challenges facing future development of the concept.

Reviewer #2 (Remarks to the Author):

The authors have addressed all of this reviewer's comments.

Reviewer #3 (Remarks to the Author):

All of my concerns have been answered. The manuscript is acceptable for publication.

Authors' response for Zhao et al., manuscript ID NCOMMS-20-45671A. Detailed response to reviewers' comments. Reviewers' comments are in black and the responses are in blue.

Reviewer #1 (Remarks to the Author):

In my opinion, the authors have substantially improved the quality of the manuscript in revision, and the result seems well suited for publication in Nature Communications, subject to minor revisions as noted below.

>> We thank the reviewer for the support of our work. In response to the reviewer's comments, we have rewritten the related part of the paper. Below we will address the reviewer's suggestions point by point.

I would suggest minor revisions to the closing paragraph on pages 14/15. In particular, I think the authors should clarify a bit more what is meant by "electric power network" (L335)... is that intended to mean power electronics currently addressed by switching power converters, e.g., consumer power supplies, chargers, etc. (say, Ws to kW), or is this meant to imply transmission and distribution applications within the power grid (MWs?) While there are interesting questions as to the technoeconomic viability of photonic transformer at increasing scale, I don't think that needs to be addressed here. My comment is motivated simply by the fact that "electric power network" seems to have a really vague meaning.

>> In light of the potential high power density, we think the photonic transformer not only can address the Ws to kW power levels for consumer power supplies, but also can be scaled up to MW level that is relevant to the power transmission and distribution. We have added some text to clarify this point. Now the sentence reads:

"While the initial application of the photonic transformer concept is likely in low power electronic circuits (Ws to kW level), one may envision that this concept can be scaled up to a power level relevant for electric power network (MW level)."

Also, I have never heard the term "fabric" used to describe a circuit topology in this context (L329). Is this intended to describe a specific switching configuration, or an additional material/layer between the solar cells and the LEDs that could otherwise change the behavior? Ultimately I feel that a few more sentences should be added to describe the capabilities afforded by the proposed approaches, in terms of voltage/current adjustment range (e.g., discrete, rectangular ratio steps, or continuous variability). Without further conceptual development, it's not clear to me that electrical switching networks alone would be capable of allowing arbitrary LED-to-PV ratios without impacting the efficiency of the device. I'm not claiming that the authors need to present any more conceptual developments in the present manuscript, just more clearly state the capabilities and limitations of the proposed approaches, and the challenges facing future development of the concept.

>> We thank the reviewer for the comment and feedback. The term ‘fabric’ was indeed used to describe a switch network that realizes adjustable voltage conversion ratio and/or output voltage; such a switch network adjusts the output voltage/current in roughly discrete, rectangular steps. In light of the reviewer’s comment, we have decided to use the term ‘switch network’ instead; the term should be more appropriate and less likely to confuse as the term is already used in other contexts to describe a similar concept such as in FPGA. We have modified the sentence which now reads:

“The conversion ratio and/or the output voltage can be modified in real time by a switch network that reconfigures the connections among the LEDs and another switch network on the PV side such that the output voltage and current can be adjusted in discrete steps.”

Meanwhile, we think it is possible to design the photonic transformer circuit in which such adjustments have little impact on the device’s conversion efficiency. For example, one can design a photonic transformer out of a 1-LED-to-1-PV building block and devise a switching strategy that switches both the LED and PV sides in lockstep so that little to no photon loss results in the adjustment process, thus preserving the device efficiency.

Reviewer #2 (Remarks to the Author):

The authors have addressed all of this reviewer’s comments.

Reviewer #3 (Remarks to the Author):

All of my concerns have been answered. The manuscript is acceptable for publication.

>> We thank Reviewers 2 and 3 for their support.